# Understanding implementation of a complex intervention in a stroke rehabilitation research trial: A qualitative evaluation using Normalisation Process Theory

Louise Johnson[1,2]*, Julia Mardo[3], Sara Demain[2]

1 University Hospitals Dorset NHS Foundation Trust, Castle Lane East, Bournemouth, Dorset, United Kingdom, 2 School of Health Sciences, Faculty of Environmental and Life Sciences, University of Southampton, Southampton, United Kingdom, 3 Dorset Healthcare NHS Foundation Trust, Yeatman Hospital, Hospital Lane, Sherborne, Dorset, United Kingdom

* Louise.Johnson@uhd.nhs.uk

## Abstract

**Data Availability Statement:** All relevant data are within the paper and its Supporting information files.

### Background

The Implicit Learning in Stroke study was a pilot cluster randomised controlled trial, investigating the use of different motor learning strategies in acute stroke rehabilitation. Participating Stroke Units (n = 8) were from the South East/West regions of the UK, with the experimental intervention (implicit learning) being delivered by clinical teams. It required therapists to change how they gave instructions and feedback to patients during rehabilitation. This paper reports the processes underpinning implementation of the implicit learning intervention. The evaluation aimed to i) understand how therapists made sense of, engaged with and interpreted the effects of the intervention; ii) compare this to the experience reported by patients; iii) extrapolate learning of broader relevance to the design and conduct of research involving complex interventions in stroke rehabilitation.

### Methods

Qualitative evaluation, with data collected through focus groups with clinical staff (n = 20) and semi structured interviews with people with stroke (n = 19). Mixed inductive and theory driven analysis, underpinned by Normalisation Process Theory.

### Results

How therapists made sense of and experienced the intervention impacted how it was implemented. The intervention was delivered by individual therapists, and was influenced by their individual values, beliefs and concerns. However, how teams worked together to build a shared (team) understanding, also played a key role. Teams with a more "flexible" interpretation, reported the view that the intervention could have benefits in a wide range of scenarios. Those with a more fixed, "rule based" interpretation, found it harder to implement, and perceived the benefits to be more limited. Therapists' concerns that the intervention may

**Funding:** This project is funded by the National Institute of Health Research (NIHR) [ICA-CL-2017-03-011]. Funding was awarded to LJ, as part of a Clinical Lectureship. The views expressed are those of the author(s) and not necessarily those of the NHS, the NIHR, or the Department of Health and Social Care. The funders had no role in study design, data collection and analysis, decision to publish, or preparation of the manuscript. www.nihr.ac.uk.

**Competing interests:** The authors have declared that no competing interests exist.

**Abbreviations:** IMPS, Implicit Learning in Stroke; NPT, Normalisation Process Theory.

impair therapeutic relationships and patient learning were not reflected in how patients experienced it.

## Conclusions

Changing practice, whether in a research study or in the "real world", is complex. Understanding the process of implementation is crucial to effective research delivery. Implementation frameworks facilitate understanding, and subsequently the systematic and iterative development of strategies for this to be addressed. How teams (rather than individuals) work together is central to how complex interventions are understood and implemented. It is possible that new complex interventions work best in contexts where there are 'flexible' cultures. Researchers should consider, and potentially measure this, before they can effectively implement and evaluate an intervention.

## Trial registration

Clinical Trials - NCT03792126.

## Introduction

The Implicit Learning in Stroke Study (IMPS) was a mixed methods, pilot cluster-randomised controlled trial, investigating the impact of implicit learning principles on the recovery of lower limb motor function in sub-acute stroke rehabilitation [1]. Implicit learning targets the non-conscious attributes of the motor learning process, [2] leading to learning without awareness. This differs from "usual care" rehabilitation, which is known to be explicit in nature, i.e. therapists typically use frequent, detailed, body focussed instructions and feedback [3, 4].

The IMPS study involved eight acute stroke units (SU's) in the UK, who were cluster randomised to either adopt principles of implicit learning during rehabilitation (n = 4), or continue with usual care (n = 4). Here we report findings from a nested qualitative study, which sought to understand therapist perceptions of implementing, and patient experiences of receiving, the implicit learning approach.

The challenge of implementing research findings within stroke rehabilitation is well documented [5–8]. Reasons for this are multifaceted, and are influenced by the complex nature of most rehabilitation interventions–which typically involve various components, require expertise to deliver, and must be flexible enough to meet the specific needs of the individual [9]. Few stroke rehabilitation studies have used implementation science theories to comprehensively understand these factors during trial delivery, limiting the potential to support effective knowledge translation into practice. Improving understanding of the factors impacting implementation at this pilot stage will ensure the intervention is developed to be workable in practice, and enable robust design of a future definitive trial.

Normalisation Process Theory (NPT), which identifies, characterises and explains key mechanisms that motivate and shape implementation, was used to underpin this research [10, 11]. NPT explains how people make sense of the work of implementing and integrating a complex intervention (coherence); how they engage with it (cognitive participation); enact it (collective action); and appraise its effects (reflexive monitoring) [11]. NPT has been used widely to describe and explain the implementation of complex interventions that already have a research evidence base [12, 13], including in stroke rehabilitation [14–18]. Here, we use NPT

to explore how a research intervention is experienced by those delivering it and undergoing it, to understand the impact of this on trial fidelity.

The purposes of this qualitative evaluation were to: i) understand how therapists made sense of, engaged with and interpreted the effects of the IMPS intervention; ii) compare this to the experience reported by patients; iii) inform future research design and potential translation into practice; and iv) extrapolate learning of broader relevance to the design and conduct of research involving complex interventions in stroke rehabilitation.

## Materials and methods

### Study design

In the pilot cRCT [1], therapists delivering implicit learning received training (2.5 hours) and a handbook containing information on the theory and principles of implicit learning, and examples for how this could be applied in practice (including images and video). The principles of implicit learning, as applied in this research, involved: a) reducing the frequency of verbal instructions during therapy sessions; b) increasing the specificity of feedback, and c) adapting tasks *and* communication to promote an external focus of attention [19]. The intervention was not prescriptive; therapists were asked to adapt their practice for the individual patient(s) they worked with, whilst staying true to these principles. The full treatment guidance is available in S1 File.

This paper reports findings from the nested qualitative evaluation. Focus groups with trial therapists at intervention sites explored implementation of the IMPS intervention. Focus groups were chosen to enable the intervention teams to collectively discuss and debate, generating rich details of the complex experiences and reasoning behind actions, beliefs, perceptions and attitudes [20]. Semi-structured interviews explored patients' experience of therapy in both the intervention and the control groups, enabling comparison between study arms, and with the perceptions of therapists. Data collected through both focus groups and interviews were considered collectively to meet study aims.

**Setting and subjects.** Focus groups (n = 4), one at each intervention site, were conducted with a convenience sample of participating physiotherapists, occupational therapists and therapy assistants. They took place at the end of the trial, when therapists had between 3 and 15 months experience of using the intervention. Due to the COVID-19 pandemic, the time period between the recruitment phase of the cluster trial, and focus group data collection, ranged between 1–12 months.

Semi-structured interviews were conducted with a convenience sample of patient participants. Interviews took place within 7 days of completion of trial interventions; ensuring participants had recent experience of rehabilitation to draw on.

Ethical approval was obtained from South Central–Berkshire B Research Ethics Committee (Ref: 18/SC/0582)

### Procedure and analysis

**Focus groups.** All intervention sites were invited, and agreed, to participate. Individual therapists and therapy assistants were provided with information about this aspect of the study. Those agreeing to participate provided written informed consent.

Face to face focus groups were facilitated by the lead researcher (LJ). Groups lasted approximately one hour, and followed a topic guide, broadly structured around the NPT framework (S2 File). Groups were audio recorded, and transcribed verbatim. An observer recorded nonverbal aspects of the discussion.

Focus group transcripts were analysed by two researchers (LJ and JM)–allowing for sense checking of ideas, and the exploration of multiple assumptions and interpretations, rather

than for consensus of meaning. Following data familiarisation, a hybrid approach [21] of both inductive and theory driven analysis was used. Both researchers independently coded the first transcript inductively, then met to discuss and agree an initial set of semantic codes; allowing identification of the explicit or surface meaning of the data, and describing the content as communicated by the respondents [22]. This was used to formulate a coding framework. The two researchers then independently applied the agreed codes to a second transcript and met again to discuss any new codes.

Data then underwent theory-driven analysis, guided by the four main constructs of NPT [23]. The semantic codes were categorised and tabulated according to the most relevant construct, as per Coupe et al. [24]. Transcripts were revisited multiple times, moving back and forth between deductive and inductive coding. Where hidden meanings or underlying assumptions were identified, latent codes were applied. This enabled analysis to move beyond a purely descriptive approach [25], through the identification of ideas or ideologies that could inform interpretation of the descriptive (or semantic) data [22]. This mixed data analysis generated a rich and detailed understanding of the underpinning meaning of the data. Once the final coding framework was agreed, the primary researcher (LJ) revisited all four transcripts to re-label data, and to extract and collate findings.

**Interviews.** Patient participants were approached by a local clinician, who provided written information about the purpose of the interviews. Those willing to participate discussed the study with the lead researcher (LJ). All research participants provided informed written consent to take part, including consent for the use of anonymised quotes in study publications.

Face to face interviews took place, in hospital (n = 17) or in the patient's home (n = 2), no later than 7 days post-intervention. A topic guide was used to structure the discussion (S1 File). Interviews were audio recorded, and reflective notes made and incorporated into the analysis [26].

Interview analysis was conducted by the lead researcher (LJ), using a thematic approach as outlined by Braun and Clark (2021) [27]. Analysis took an experiential orientation [22], prioritising how each style of therapy (implicit learning or usual care) was experienced by the participant. Initial coding was predominantly inductive; codes were transferred onto a spreadsheet, with a label identifying the participant and group (intervention or control). Codes were then considered collectively, to interpret meaning and meaningfulness across the dataset. At this stage, connections were also made to the focus group themes; with interview (patient) data being broadly mapped to the focus group (clinician) data. This process of reviewing and comparing the two datasets underwent several iterations, informed by discussion with the wider study team (SD and JM).

Here, we report aspects of the patient data that we could 'cross-reference' to focus group themes, enabling us to understand what therapists *thought* might be occurring for patients, and how patients actually experienced this. As part of the analytic process, interview and focus group data were discussed for reflective purposes with a third researcher (SD).

## Results

Tables 1 and 2 provide demographic details of the 20 therapists attending the four focus groups, and the 19 patients interviewed (control group n = 10; intervention group n = 9). A summary of codes, and how these relate to each NPT construct, is given in Table 3.

Focus group participants were from four centres, each of which were similar in terms of rehabilitation context (all provided hyperacute and acute stroke care), and unit size.

Findings are presented below according to each of the four NPT constructs, and how they are represented through participant views and experiences.

**Table 1. Focus group—Participant characteristics (Clinicians).**

|  | Site 2 * | Site 5 | Site 6 | Site 7 |
|---|---|---|---|---|
| **Number Invited to Participate** | 6 | 7 | 5 | 5 |
| **Number of Participants** | 5 | 6 | 4 | 5 |
| **Profession (n)** |  |  |  |  |
| Physiotherapist | 5 | 5 | 3 | 3 |
| Occupational Therapist | 0 | 0 | 1 | 0 |
| Therapy Assistant | 0 | 1 | 0 | 2 |
| **Male: Female** | 1:4 | 0:6 | 2:2 | 2:3 |
| **Experience**–years working in stroke rehabilitation (range) | < 1–15+ | 1–15+ | 3–15+ | 1–15+ |

Focus groups were conducted at the four intervention sites that were enrolled in the trial.

* 8 sites took part in the trial, with four randomised to intervention. As this qualitative study was about experiences of implementing the intervention, only those sites randomised to the intervention arm were invited to take part.

## Coherence—Professional values

*Coherence refers to the sense-making work that people and teams do, when they are faced with the challenge of operationalising a new set of practices*

[23].

**Table 2. Interview—Participant characteristics (Patients).**

|  |  | Intervention Group | Control Group** |
|---|---|---|---|
| Type of Stroke: |  |  |  |
|  | TACS | 0 | 0 |
|  | PACS | 6 | 2 |
|  | POCS | 1 | 1 |
|  | LACS | 2 | 3 |
|  | ICH | 0 | 4 |
| Time Since Stroke (at enrolment to trial)–days Median (IQR) |  | 6 (3–12) | 5 (3–12) |
| Age–years Median (IQR) |  | 66 (45–94) | 76 (25–87) |
| Gender–n Male: Female |  | 3:6 | 6:4 |
| Stroke Severity—NIHSS on admission Median (IQR) |  | 8 (4–14) | 8.5 (2–16) |
| Cognition–MoCA Median (IQR) |  | 24 (20–27) | 20 (12–27) |
| Language Deficit Yes: No |  | 1:8 | 0:10 |

MoCA–Montreal Cognitive Assessment; NIHSS–National Institute for Health Stroke Scale;

TACS–total anterior circulatory stroke; PACS–partial anterior circulatory stroke; POCS–posterior circulatory stroke; LACS–lacunar stroke; ICH–intra cerebral haemorrhage.

** for the purpose of understanding and describing experiences of the IMPS intervention, only data from the intervention group is used. For the purpose of comparing experiences between the two styles of therapy delivery, data from both intervention and control groups is used.

**Table 3. Coding framework and descriptors (Focus group).**

| NPT Construct | NPT Component | Code | Descriptors/exemplars |
|---|---|---|---|
| **COHERENCE** | **1. Differentiation** understand how a set of practices and their objects are different from each other | a) Theoretical Understanding—of the intervention | • Understanding of the core components of the approach. <br>• Description of how these components were contextualised. <br>• Interpretation of specific components–e.g. handling, task specificity. |
| | | b) Different from usual practice | • Examples of participants confirming ILA to be different to their usual practice. <br>• Reinforced by other themes–requires time to think/plan; difficult to do. |
| | | c) Perceived ease of application | • Judgements (in early coherence phase) on how difficult (or not) it might be to apply the approach; and in what contexts. <br>• Reflection on how this might feel (e.g. reducing communication). |
| | **2. Communal Specification** people working together to form a shared understanding of the aims, objectives, benefits or a set of practices | d) Shared Understanding | • Reported activity that would support communal specification, e.g. <br> • Sharing ideas at service level—sharing ideas about the practical application of the approach <br> • Sharing ideas at patient level—team treatment planning for individual patients |
| | | e) Social Influence/ Opportunity | • More likely to adopt an approach if others around you practice in a similar way. <br>• Positive culture towards research; changing practice; different approaches. |
| | **3. Individual Specification** things people do to help them understand their specific tasks and responsibilities around a set of practices | f) Principle Application | • Examples of work done by teams to support the application of principles, e.g. time spent reflecting on treatment application, problem solving. |
| | **4. Internalization** understanding the value, benefits and importance of a set of practices | g) Constructing an idea of (potential) value/benefits | • Judgements about who the approach might work well for, and where other approaches might be better. |
| | | h) Incongruous priorities (therapist-patient) | • Acknowledging that there are differences between therapist and patient views, e.g. in evaluation of movement |
| | | i) Professional values | • How the approach fits with past experience <br>• How the approach fits with underpinning values, e.g. in relation to the role of communication <br>• Therapeutic relationships |
| | | j) Applicability in different contexts to different patient groups | • Clinical scenarios (patient characteristics or activity characteristics) where therapists would deem the approach to be more or less applicable. <br> • Patient characteristics <br> • Activity characteristics |

(*Continued*)

**Table 3.** (Continued)

| NPT Construct | NPT Component | Code | Descriptors/exemplars |
|---|---|---|---|
| **COGNITIVE PARTICIPATION** | **Initiation**<br>are key participants working to drive a set of new/ modified practices forward | k) Endeavour | • Examples of teams wanting to "rise to the challenge"–keenness to give it a go.<br>• Individuals or teams with positive/constructive attitude towards the research–they see difficulties but are looking for solutions. |
| | | l) Collective action | • Examples of teams/individuals taking action to move implementation forward |
| | | m) Opinion leaders | • Champions and their influence |
| | **Enrolment**<br>participants may need to organise/re-organise themselves in order to collectively contribute to the work involved; rethinking individual and group relationships between people and things | n) Practical enablers | • Organisation of space and equipment |
| | **Legitimation**<br>work of ensuring that other participants believe it is right for them to be involved, and that they can make a valid contribution | This component related to the capacity and willingness of individuals to organise themselves in order to collectively buy into the intervention. We did not identify themes that fit within this specific component. It is potentially less relevant as our work was a research project, and participants were therefore following a research protocol with pre-assigned roles. | |
| | **Activation**<br>collectively define the actions and procedures needed to sustain a practice and stay involved | o) Sustainability enablers | • True implementation needs the development of expertise.<br>• Becomes easier to apply and to adapt with practice.<br>• Training opportunities. |
| | | p) Intention to continue | • Examples where participants reported translating/ integrating principles into practice (with non-research patients)<br>• Individuals and/or teams demonstrating the intention to continue to use what they have learned as part of their normal practice. |

*(Continued)*

**Table 3.** (Continued)

| NPT Construct | NPT Component | Code | Descriptors/exemplars |
|---|---|---|---|
| **COLLECTIVE ACTION** | **Interactional Workability**<br>interactional work that people do with each other, with artefacts, and with other elements of a set of practices, when they seek to operationalize them in everyday settings | q) Operationalising principles in specific context | • Therapists compartmentalising the therapy aspect of rehabilitation–applying the principles in a controlled space. |
| | | r) Use of equipment | • Using "kit" as an enabler–therapists tried to incorporate the basic kit into their sessions. When doing this, felt more able to apply IMPS principles–because it was something tangible and different. |
| | | s) Challenge in application (experienced) | • Harder (for the therapist) to achieve the desired movement.<br>• Challenging not to use internally focussed terminology<br>• Reporting it to be more challenging to apply in certain patient groups. |
| | | t) Ease in application (experienced) | • Reporting it to be easier to apply with certain patient groups. |
| | **Relational Integration**<br>the knowledge work that people do to build accountability and maintain confidence in a set of practices and in each other, as they use them | u) External validation | • Therapists conscious of how treatment session appears to others (patient, carers, other members of the MDT) |
| | | v) Broad purpose of communication | • Communication as an inherent part of physiotherapy–perceptions on the impact of changing communication (e.g. on therapeutic relationships) |
| | | w) Patient perceptions/ experience | • Therapist thoughts on patient experience, e.g. in recognising progress. |
| | **Skill Set Workability**<br>the allocation of work that underpins the division of labour that is built up around a set of practices as they are operationalized in the real world | All sites recognised that they were building their skill set. In terms of implementation of the approach, they didn't divide tasks, but some centres deliberately enabled all staff to work with study patients, and in others it tended to be one or two senior staff. Overall, this component was less relevant as the participants were following a research protocol, and therefore had less scope to allocate "work". | |
| | **Contextual Integration**<br>the resource work–managing a set of practices through the allocation of different kinds of resources and the execution of protocols, policies and procedures | x) Resource requirements | • Examples of resources that participants perceived would aid integration–e.g. broader exercise toolkit, progression/ regression examples. |
| | | y) Wider MDT | • Recognition that to be beneficial, likely to require wider implementation (i.e. over 24 hour period)<br>• Perceived as likely to be very difficult to change nursing practice to align with the approach–report that nursing practice is typically explicit.<br>• Therefore question value if only applied during therapy. |

(*Continued*)

**Table 3.** (Continued)

| NPT Construct | NPT Component | Code | Descriptors/exemplars |
|---|---|---|---|
| **REFLEXIVE MONITORING** | **Systemisation** participants may seek to determine how effective and useful it is for them and for others, and this involves the work of collecting information on a variety of ways | z) Evaluation (of outcome / benefit) | • Informal methods used to evaluate benefit—observation of movement; judgements about progress/success. <br> • Therapists sense of value of the intervention |
| | | aa) Alignment with other treatment aims | • Perceived secondary benefits that fit with other treatment aims, for example increased intensity of practice; increased patient autonomy. |
| | **Communal Appraisal** participants work together, sometimes in formal collaborative, sometimes informal groups, to evaluate the worth of a set of practices. They may use many different means to do this, drawing on a variety of experiential and systemised information. | bb) Informal discussion about approach | • Examples of teams working together to reflect on and/or evaluate the benefits. |
| | **Individual Appraisal** participants work experientially as individuals to appraise the effects on them and the contexts in which they are set. From this work stem actions through which individuals express their personal relationships to new technologies or complex interventions. | cc) Experiential learning / development | • Learning about the approach prompted therapists to be more creative. |
| | | dd) Reflect on prior knowledge of approach (or lack of) | • Stimulated critique of usual care |
| | **Reconfiguration** appraisal work by individuals or groups may lead to attempts to redefine procedures or modify practices, and even change the shape of a new technology itself. | ee) Reconfiguration (hypothetical) | • Examples where therapists discussed how they might adapt the approach to further implementation, or what they might need to know in order to do this. |

All groups indicated their interest in the concept of implicit learning. For some, the principles aligned to the value they place on evidence-based practice, and their own understanding of motor learning.

> *"I think when I first heard about it I was really excited. It really feels like it [implicit learning] fits well with my values about working as a therapist, and my experience of what works"*

[Physiotherapist, Site 5, Participant 2]

Some suggested the approach gave patients more control and motivation, viewing this as a good thing:

> *". . .in my experience of seeing patients and how quickly they become institutionalised and how they lose control and direction of what's happening to them so rapidly. . . it just disappears away from them and it [implicit learning] felt such an opportunity for exploring how to give that control back to people*

[Physiotherapist, Site 5, Participant 1]

An opposing view was that reducing and altering verbal communication might negatively impact patient-therapist relationships, compromising motivation, rapport, and trust. Several participants reported that silence could feel uncomfortable, although they were unclear for whom (themselves or the patient).

> *". . .it's was hard. . ...you can still give positive feedback, but not saying 'well done' is quite hard to hold yourself back, because you want to be encouraging, and limiting how much you are talking feels like you are just not caring"*

[Physiotherapist, Site 2, Participant 3]

These therapist-driven values were at odds with the experience of patients, who were consistently positive about their therapy, with no negative perceptions of communication style. Whilst therapists described satisfaction a being seen as "kind", patients placed value on a broader set of characteristics, including sessions being "*structured*", "*purposeful*", "*challenging*" and "*educational*", and of therapists as those with the expertise to help their recovery. Although the use of motivational praise was deliberately less frequent in the implicit learning group, patients in both groups highlighted that positive reinforcement was motivating and reassuring; reducing its frequency did not appear to have negative consequences.

"*They [therapists] tell you that you are doing well. They've always been very positive with me. It's helpful*"

[Patient Participant, 5-F, Intervention]

Several patients receiving the implicit learning intervention commented on the benefit of clear and concise communication during rehabilitation; and the value of measurable indicators of progress.

"*They only tell you the things you need to do, they don't waffle around it, there's no chit chat there to confuse you, and it's exactly what you need to do, focus on it, move on. . .I find* [that approach] *the best way, I'd rather it.*"

[Patient Participant, 6-B, Intervention]

"*. . .and I put my weight on my leg for 5 minutes more. I think that session I was timed at 5 minutes with equal weight on both feet. We did one session with electronic scales. . ..which was a good way to know which leg was doing the work*"

[Patient Participant, 5-B, Intervention]

These views were not heard from patients in the usual care group.

Therapists perceived that patients valued their professional feedback on movement; some believed this could only be communicated explicitly. This view of the therapist as the professional expert was also reflected by patients, but did not appear to relate to communication style.

"*The therapists guide me, so they know what process to do. I tell them if there is a problem, if I'm not doing this or I can't do that–they listen and they try to correct it. I think you are working as a team really, that's the way I look at it. They're the instructors and I'm the novice*"

[Patient Participant, 2-I, Intervention]

**Coherence—Differentiation and theoretical understanding.** Therapists provided multiple examples confirming the implicit learning approach as different from their usual practice (differentiation). No-one indicated that their usual approach was predominately implicit.

"*I'd heard of [implicit learning], yes, just when you read about different principles of learning and things like that, and how we learn stuff, but never really applied it.*"

*[Physiotherapist, Site 2, Participant 2]*

Therapists understood the core components of implicit learning, but demonstrated differences in the degree to which they understood the underpinning motor learning theory. This resulted in variances in how they contextualised the approach, and their interpretation of its practical application. These differences appeared to be created within teams, rather than by individuals. Broadly speaking, teams understood the approach to be either a set of:

- *principles*, that can be applied flexibly; i.e. implicit and explicit learning is part of a continuum; therapy can be adapted to fit at different places within that continuum; *or,*

- *rules*, which are fixed and discrete; i.e. a therapy intervention can be either implicit or explicit

Teams (n = 2) that reported working together in an organised and collective way to understand the approach (communal specification) adopted the "flexible view". It was within these teams that we noted greater and richer discussion. Therapists were more likely to discuss, critique, and reflect on the approach if they did so collectively. Through doing so, they built a greater theoretical understanding, and a more positive and solution focussed approach to application, which they were able to apply in a range of scenarios.

> *"It was a real conversation generator wasn't it, in how we were going to deliver the different intervention and make sure that we were trying to adhere to the structure of delivering therapy in a different way."*

*[Physiotherapist, Site 5, Participant 2]*

Conversely, in focus groups reporting a more "rules based" approach, we observed less dynamic conversation and debate, and the views of clinical leaders typically dominated the discussion. These "rules based" teams (n = 2), described implicit learning as less coherent with their views of 'good' therapy, particularly in relation to handling "handling" is as general term used in neurological rehabilitation to describe the provision of manual support from a therapist(s), during movement. This may be to ensure patient safety, to provide feedback (e.g. sensory), or to guide and facilitate movement., movement quality and functional task practice.

Handling is not inherently counter to an implicit approach, and there was nothing within the training or treatment guidance relating to this; however, most therapists saw the implicit approach as being "hands-off". Handling was viewed by some as a separate therapeutic approach, and individuals reflected that they didn't know how to incorporate handling alongside implicit principles. This was a contentious issue within the "rules based" groups, where clinicians placed high value on the importance of handling; and felt that this was compromised by implicit learning principles. These views were not challenged or debated more widely within those groups.

> *"I thought as soon as you put your hands on to that person that it's becoming more explicit, but I kind of thought we were aiming to be hands off as much as possible, but if we had to be hands on we could a little bit but I didn't think we could support the patient and things like that which is why I found it very difficult moving it into gait"*

*[Physiotherapist, Site 2, Participant 1]*

Movement "quality" was also raised as a specific challenge. Therapists found it difficult to achieve the desired movements through adjusting a task, and preferred to correct movements

verbally and manually. Therapists also reflected that if movement quality was compromised in order to enable more independent practice, this would "*not look good*" to others. Interestingly, patients also commented on the quality of movement, and of needing to "*get it right*", but they did not perceive movement quality to have the same importance as therapists.

> "*I was not lifting my foot properly, like it was dragging a little bit on the ground when you are walking,* [they instructed me to] *lift it up more, things like that. When you first start, it's not pretty. It's not dainty. It's not anything, and actually it is still not brilliant now, but it is getting better, it's just practice*"

> [Patient Participant, 6-A, Intervention]

Therapists also discussed the importance of making therapy functional, but held conflicting views on how 'function' sits within the explicit-implicit continuum. Some viewed a functional approach as closely aligned to implicit learning; others found it hard to apply the principles in a functional context. Two focus groups indicated that it was difficult to meet their therapeutic aim of making all tasks functional using an implicit approach, but could not illuminate how this would be easier with an explicit approach.

## Cognitive participation—Endeavour and experiential learning

> *Cognitive participation is the relational work that people do to build and sustain a community of practice, around a new intervention*

> [23].

All groups recognised that applying implicit learning required a new skill set, and that true implementation required expertise. Implicit principles became easier to apply and adapt with experience. Deeper understanding of the approach may come with greater exposure, but the opportunity for advanced training was suggested as a necessary enabler for sustainability. Interestingly, the more therapists tried to work with the approach, the more they realised that it required expertise to integrate into practice–something that hadn't been anticipated at the outset.

> "*I think when we understood it [at the beginning] I think everybody thought that we sort of did elements of it already. But actually when we got down to delivering it and trying to do it as true to what we were trying to achieve in the research then it was harder to plan*"

> [Physiotherapist, Site 6, Participant 2]

Teams who expressed more positive opinions of the implicit approach also reported a greater willingness to initiate a change in their practice; this seemed to arise from genuine intrigue and interest in trying to change their behaviour. This endeavour–wanting to "rise to a challenge"–influenced how that team subsequently engaged with the intervention. All teams identified similar challenges, but those demonstrating 'endeavour', also reported a more flexible interpretation of the approach and greater ability to find solutions to its delivery.

As part of developing their understanding of the approach, therapists made judgements about how difficult it might be to apply, and how this may differ in various scenarios. Unanimously, therapists recognised that usual practice included a high level of communication, and anticipated this would be difficult to change.

*"I think the [usual] focus on body parts was something that stuck out probably for all of us, just in terms of how difficult that was going to be. . . .I recognised that that was going to be a difficulty to apply straight away because we're aware of how much we talk about different parts of the body when we treat people."*

[Physiotherapist, Site 2, Participant 1]

*"And just limiting our language in general like limiting how much you say to someone. . . . so I just thought I don't know how I'm going to do that. You feel like you really need to speak."*

[Physiotherapist, Site 5, Participant 3]

Therapists also identified different clinical scenarios, relating to characteristics of patients or activities—where they anticipated that the approach may be more or less valuable. There were multiple examples of contradictory views between focus groups. For example, there were opposing views on the potential benefits of implicit learning in patients with communication and cognitive impairments, and in sitting balance and gait tasks.

"*I think* [explicit learning] *is what* [people with cognitive or communication impairments] *need, because they need more commands or more demonstration to follow what we want them to do, compared to those who are very good with the cognitive and expressive things..*"

[Physiotherapist, Site 7, Participant 1]

"*With people with cognitive impairments they don't want that cognitive overload, they don't want people shouting instructions at them from all sides at the same time. I think it can be better to have more implicit therapy.*"

[Occupational Therapist, Site 6, Participant 3]

Finally, some therapists discussed how they had integrated principles into their wider practice (i.e with non-research participants)–because they perceived it would be beneficial. This "activation", an intention to continue to use what they had learnt, indicates the potential for implicit learning principles to be integrated within usual care.

*"I think it still takes me a lot more thought but I've really enjoyed it and I think keep trying to use it but it does take that planning and I'm always trying to think what am I saying, should I be saying that, have I given too much."*

[Physiotherapist, Site 5, Participant 3]

## Collective action

*Collective action is the operational work that people do to enact a set of practices*

[23].

Participants reported differing experiences of operationalising implicit learning principles. Designing and adapting interventions to achieve the desired movement outcome required more thought and planning, as did knowing how to progress or regress interventions. Some saw this as a barrier, whilst others recognised this as an inevitable challenge of something new that would improve through practice.

*". . ..sometimes I couldn't seem to relay the information I wanted, I didn't get the result I wanted from the patient. . .sometimes it was just easier to assist them to get there but then it didn't necessarily, sometimes it worked and sometimes it didn't"*

[Physiotherapist, Site 7, Participant 5]

*"I think you would probably develop a bank of activities and progressions and regressions that you probably end up using more frequently, and it will become more slick I would have thought. I'm still trying to build up that box of ideas I guess, but it'll come"*

[Physiotherapist, Site 5, Participant 2]

These differences appeared to partly relate to how far this approach deviated from individual therapist's preferred therapeutic style. How readily teams worked together to share ideas and problem solve also influenced perceived "ease" of implementation, this culture of team engagement varying between services. Therapists described feeling conscious of how a treatment session might appear to others–particularly professional peers. This seeking of "*external validation*" related to how a session might *look* and/or how it might *sound*. They also recognised that the high levels of communication seen within standard care are largely habitual, and reflected that this may have developed as students when they were required to demonstrate their knowledge and reasoning to clinical educators. This desire to demonstrate expertise, whether to patients or colleagues, was part of their identity.

*"I also wonder if there's something about how a treatment session sounds and I sometimes think about that* [what other people think] *in treatment sessions where there's often a lot of silence and not talking."*

[Physiotherapist, Site 2, Participant 2]

Whilst activities to enact the intervention were primarily dependent on changing behaviour of therapists, we noted an overall lack of autonomy that patients held in this process. When asked about their rehabilitation sessions, patient participants found it difficult to describe what they had been working on, or why.

*". . .I mean they're trained, I just do what they say. They're trained, that is their profession. . .. you don't want to let that guy or that girl down. You just want to please them."*

[Patient Participant, 3-D, Usual Care]

## Reflexive monitoring

*Reflexive monitoring is the appraisal work that people do to asses and understand the ways that new set of practices affect them, and others around them*

[23].

Therapists made judgements about the effectiveness of the intervention in different contexts, using informal methods to inform this view, such as observation of movement and evaluation of progress toward a goal. Progressing through perceived stages of recovery, and working toward a goal, was also important to patient participants.

*". . .and where I was likely to get to, or where it might be possible to get to and how they [therapy team] were going to achieve that. That makes a big difference, having that focus and goal0"*

[Patient Participant 6-B, Intervention]

Individually, and collectively, therapists reported how involvement in the study had stimulated discussion and critique of their communication practices, something that they did not typically consider. Despite the challenges and different views on application, there was collective recognition that 'usual care' may not always be most effective.

*"It is, it's like we can't shut up. I genuinely think it is we are hardwired to narrate what it is we're doing with patients all the time. I don't know if it's for our benefit or theirs to be honest."*

[Occupational Therapist, Site 6, Participant 3]

*"So I think we definitely used a lot more body terminology before using the approach and definitely said 'well-done' far too much. . .every word you are saying actually counts a little bit more and it's making the patient think about achieving a movement rather than thinking of all the individual sections of that movement which then don't actually end up creating a good movement pattern anyway"*

[Physiotherapist, Site 2, Participant 2].

Repeatedly, the terms "ease of applying" and "applicability" were used interchangeably. The approach was perceived as beneficial if it was easy (for the therapists) to apply, and the wrong approach if it was harder (for the therapists) to apply. This didn't always appear to be backed by objective evidence of actual benefit/lack of benefit. Instead, effectiveness was judged on how much time and effort the therapist needed to dedicate to planning treatment sessions.

Most appraisal work was generated by discussion and reflection within teams. Within this, there were only a small number of occasions where therapists utilised the reported views or experiences of patients to inform their own professional views. There were no instances of therapists using more structured methods to understand what worked or didn't work from a patient perspective. For example, we identified differences between the groups in terms of if/how frequently patients created "rules" for movement. Rules (or methods/techniques) are statements, made by the patient, that highlight that person has used explicit information to improve or maintain their performance [28]; typically relating to the movement or position of a limb. Within the interviews, those in the control group reported rules more frequently, than those in the implicit learning group.

*"At the moment, I am practising standing up, leaning over and reaching and I have to lock my leg in and push my bottom back and sit down, but that comes from holding onto the edge of the chair. . . . . .up and standing and leaning over to the right, keeping my stomach into the bar and looking up. And trying to remember all of those things."*

[Patient 1-A, Control Group]

*"I think with this progress I make, there are some stages at the beginning where you need to concentrate a lot. . .but the next stage is just to get rid of the focus and move, unconscious, like automating. We do not think in normal life that we are walking. It's important to switch from training to just doing it, walking and singing a song, or talking to someone"*

[Patient 2-F, Intervention Group]

A reduction in the creation of rules is an indicator of successful implicit learning implementation, as it emphasizes a reduction in conscious control of movement. However, it was unclear how aware of this therapists were, and the potential role that understanding rules (through self-report from patients) could have on building or reinforcing collective action.

Some therapists discussed how they might adapt the approach to further implementation, or what they might need to know in order to do this. This reconfiguration work only came from those teams who had been flexible in how they interpreted the approach; and were therefore more readily able to think about its fit within their practice. These teams discussed who the approach might work for, and differentiated the actual treatment session from other aspects of the physiotherapists role.

*"I can imagine long term. . . .you pick 'who does this fit for'? And we'd get better and more skilled at deciding that. . . .Also, it's almost that you could separate the session and do the active stuff you are doing* [using implicit principles] *and then have an analysis afterwards."*

[Physiotherapist, Site 6, Participant 4]

## Relationships between NPT constructs

Although we have described the four NPT constructs and their components separately, there were relationships between and across themes (Fig 1).

Activities relating to coherence–the sense making work—occurred early in the process of implementation. At this stage, teams broadly fell into one of two groups, with more fixed (rule based) or more flexible (principle based) interpretations. This early formation of understanding impacted on how individuals and teams went on to enact the approach, and how they interpreted its value. The "rule-based" teams typically perceived the approach to conflict with their professional values, particularly with regards to their role as a therapist, and their relationship with patients. Teams that developed this view reported implicit learning to be more challenging to apply, and as a result perceived it to be beneficial in a narrow range of clinical contexts. This fed into the view that it conflicted with professional values, which became a barrier to implementation.

The "flexible" therapists reported more activity that aligned with communal specification (working together to form a shared understanding) and also discussed a commitment to making the approach work (endeavour). The dynamic within these focus groups was more collaborative, with senior clinicians facilitating others to contribute their views, and a more open and questioning culture. These groups appeared to find it less challenging to apply the approach; and reported a belief that it could have benefits in a wide range of contexts. These teams also developed their thinking further in terms of reflexive monitoring, reporting more in-depth communal and individual specification. Subsequently, they also reported activities that would align to (hypothetical) reconfiguration–appraising the intervention in a way that would lead them to redefine or shape it to enable sustainability in future practice

## Discussion

Using Normalisation Process Theory, we have described the process of implementing a complex rehabilitation intervention, within the context of a research trial. Understanding implementation in trials is an important aspect of feasibility and pilot work. An improved understanding of the implementation process will enhance the earlier stages of trial design, and will maximise future potential for translation of findings into clinical care.

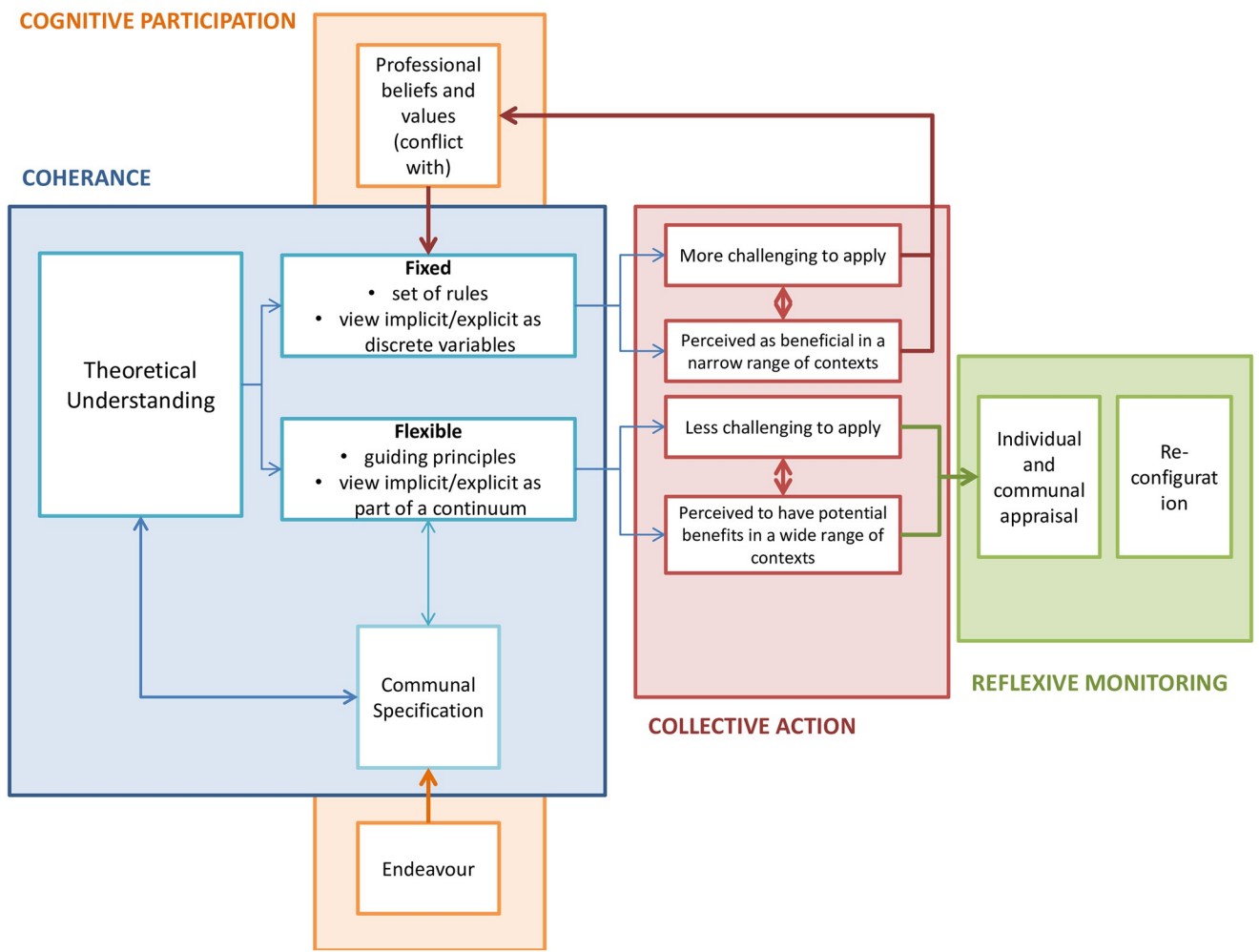

**Fig 1. Illustrates the key relationships between and across NPT themes, and how this varied between participating teams.**

We identified behaviours that mapped to all four constructs of NPT, with coherence and cognitive participation as dominant themes. We found the initial phase of coherence to be the starting point for implementation. The "work" that happened during this phase appeared to predict how subsequent stages of implementation played out. This may be due, in part, to the linear nature of research; where activities typically occur sequentially, in line with a pre-defined protocol.

## Using implementation frameworks to maximise intervention fidelity

Whilst there is recognition of the importance of measuring and reporting intervention fidelity within complex intervention trials [29–31], evidence to guide researchers on *how* to achieve fidelity is limited [32]. Proposed models typically focus on process, ensuring that: the intervention is comprehensively described; training is standardised; practitioner competence is assessed; and methods are in place for monitoring delivery [33]. Although these factors were in place in our trial, we still identified differences in how teams interpreted, engaged with, and delivered the intervention. Our findings suggest that complex rehabilitation intervention

research should consider the on-going process of implementation, not solely the end point of fidelity. Trial designs may need to actively facilitate implementation, particularly monitoring and supporting coherence and cognitive participation.

The context for this study was a randomised controlled trial in stroke rehabilitation. Implementation frameworks are typically used in both research and real-world projects to understand the process of implementation of **proven interventions** [34, 35]. They provide a structure for: describing and/or guiding the process of translating effective interventions and research evidence into practice (process frameworks); analysing what influences implementation outcomes (determinant frameworks); and evaluating implementation efforts (outcome frameworks) [36]. This study investigated implementation of, and fidelity to, an **experimental intervention**, within a research trial, demonstrating the key role of implementation frameworks in experimental research; assisting researchers to consider wider factors influencing implementation, and how these may be facilitated to optimise fidelity. For example, training delivery may need to be phased and adaptive, accounting for the fact that coherence is an on-going process, and is different between individuals and between teams. Recognising that healthcare professionals' decision making is most often informed by their interaction with each other and with opinion leaders [37], and that communal specification played an important role in how teams made sense of the intervention in our study, activities that encourage this should be built into the study design. Actively facilitating reflexive activities for teams, for example through case reviews or action learning sets, could also benefit implementation.

The importance of understanding specifically how the intervention interacts within the context in which it is implemented, is now recognised as integral to complex intervention research [9]. However, these nuanced social factors of intervention delivery are rarely considered in stroke rehabilitation trials. To deliver a new intervention, a healthcare organisation (or service) needs to be *willing* (e.g., committed) and *able* (e.g., capable) to embrace the required change [38]. Whilst there are theories and tools to aid understanding of readiness for implementation [39, 40], these are not specific to research contexts. Understanding the readiness of potential research sites to implement a new intervention could inform site selection and/or help to shape a more bespoke approach to implementation support. Developing a trial specific survey to understand a) previous attitudes towards innovation and b) readiness or intention to implement the new trial interventions, may be a useful starting point. Where potentially inflexible cultures are identified, a more tailored programme of support, informed by behaviour change theory, may be required.

## The influence of clinician values, beliefs and attitudes on implementation

Complex interventions are often reliant on changing the behaviour of clinicians. In our study, clinicians were interested in the approach and committed to participating in the research, but implementation was impacted by professional values, and attitudes toward the intervention varied. Whether teams were flexible or rule-bound was key to the extent that the intervention was accepted and implemented, and could be reflective of the wider culture within the Units. Values played a dominant role in how the intervention was perceived, understood, and subsequently delivered. The role of an "opinion leader" was evident in how teams constructed their understanding of the approach, and subsequently how they engaged with it. Specifically, the values expressed by those in clinical leadership positions influenced the views expressed by others within the group.

The issue of implementation fidelity being moderated by clinicians beliefs or concerns about interventions has been reported elsewhere in the stroke rehabilitation literature [17]. Success of implementation can be determined by the congruence or 'fit' of an intervention

with the pre-existing beliefs of those delivering it [17]. Some participants' existing beliefs inhibited delivery of the intervention. Importantly, this was not universal; there were also examples of participants who understood and adopted the intervention in an insightful way, supported by teams who were flexible and creative. Whether implementation is taking place within a research trial or clinical, practice, factors relating to attitudes and beliefs must be identified and practically addressed.

### The contrasting views of therapists and patients

Therapists discussed the significance and value of the "therapeutic relationship" they hold with patients, highlighting the importance of building rapport and creating trust in their clinical expertise. This has been reported elsewhere [41], and is likely to have a positive association with clinical outcomes [42]. In this study, therapists were asked to alter their communication style specifically during active therapy (i.e. when patients were exercising or practicing tasks; and in relation to those exercises/tasks), with the hypothesis that more concise, specific, and externally focussed communication can aid the process of motor learning. Some therapists expressed discomfort with making these changes to communication style, citing the impact they perceived it may have on therapeutic relationships, or on movement quality. Patients, however, did not express any concerns about the relationship or quality of therapy. This raises questions about whether therapists could do more to understand what type of communication works best for individual patients, how their learning preferences can be taken into account. Studies investigating intensive exercise [43] and self-directed upper limb rehabilitation [44] have also identified therapist concerns about new ways of working that are not reflected in patient experiences or outcomes. This suggests that an iterative process of implementation support may be necessary, feeding data (including patient views) back to therapists, allowing their concerns to be allayed.

### Strengths and limitations

The use of a theoretical framework to understand the process of implementation within a research study enabled the systematic and comprehensive understanding of the processes underlying implementation, meaning results are more likely to be transferable to similar contexts. As Units participating in this study were from one geographically defined region (Southern UK), it is not known if or how the findings would differ in other settings, for example those with greater social or cultural diversity. One criticism of NPT is that it focuses primarily on the action of professionals, and does not take into account a range of perspectives, including the role that end users (e.g. patients) may play in implementation [12]. This study addresses this by including data from patient interviews, demonstrating that many of therapists' concerns were not shared by patients. As our evaluation took part at the end of the trial, our analysis is limited to the activities reported by clinicians, at that point in time. Recognising that implementation is a dynamic and on-going process, evaluating this at different time points within the life cycle of a research project would provide deeper insights into the process, and how this can be maximised in future studies.

### Conclusion

The early design phase is critical in trials investigating complex interventions. The complexity of the social processes underpinning intervention delivery is greater than typically considered in stroke rehabilitation research, which mostly focuses on the 'what' and not the 'how and why'. Understanding the process of implementation is crucial, as erroneous conclusions may be drawn as to the efficacy of an intervention if insufficient thought has been given to

maximising fidelity at the trial stage, and the human and organisational processes underpinning this. How teams (rather than individuals) work together is central to how complex interventions are understood and subsequently implemented. It is possible that new complex interventions work best in contexts where there are 'flexible' cultures, and that researchers should consider, and potentially measure this, before they can effectively implement and evaluate an intervention.

## Supporting information

**S1 File. Implicit learning intervention guidance.**
(PDF)

**S2 File. Focus group topic guide.**
(DOCX)

**S3 File. Interview topic guide.**
(DOC)

**S1 Data.**
(ZIP)

## Author Contributions

**Conceptualization:** Louise Johnson, Sara Demain.

**Data curation:** Louise Johnson.

**Formal analysis:** Louise Johnson, Julia Mardo.

**Funding acquisition:** Louise Johnson.

**Investigation:** Louise Johnson.

**Methodology:** Louise Johnson, Sara Demain.

**Project administration:** Louise Johnson.

**Supervision:** Sara Demain.

**Validation:** Sara Demain.

**Writing – original draft:** Louise Johnson.

**Writing – review & editing:** Julia Mardo, Sara Demain.

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
