## [Decision Letter · Decision Letter 0]

14 Nov 2022

PONE-D-22-24727Understanding implementation of a complex intervention in a stroke rehabilitation research trial: a qualitative evaluation using Normalisation Process Theory.PLOS ONE

Dear Dr. Johnson,

Thank you for submitting your manuscript to PLOS ONE. After careful consideration, we feel that it has merit but does not fully meet PLOS ONE’s publication criteria as it currently stands. Therefore, we invite you to submit a revised version of the manuscript that addresses the points raised during the review process.

We look forward to receiving your revised manuscript.

Kind regards,

Adetayo Olorunlana, Ph.D.

Academic Editor

PLOS ONE

Journal Requirements:

Reviewers' comments:

Reviewer's Responses to Questions

**Comments to the Author**

1. Is the manuscript technically sound, and do the data support the conclusions?

Reviewer #1: Yes

Reviewer #2: Partly

2. Has the statistical analysis been performed appropriately and rigorously? 

Reviewer #1: N/A

Reviewer #2: N/A

3. Have the authors made all data underlying the findings in their manuscript fully available?

Reviewer #1: No

Reviewer #2: Yes

4. Is the manuscript presented in an intelligible fashion and written in standard English?

Reviewer #1: Yes

Reviewer #2: Yes

5. Review Comments to the Author

Reviewer #1: Congratulations on a well-written paper, presenting an example in stroke rehabilitation research how to apply NPT for implementing (and evaluating) a new intervention into practice. Outcomes of this paper strengthen the main trial outcomes as they provide information on why and how implementing implicit learning has worked and where it can be improved.

A few minor comments:

p. 10: Provide one additional sentence after Table 3 that explains to the reader that you are now presenting the NPT constructs and how they are reflected in participant statements/experiences.

p. 30, line 502: Did you assess context of the participating sites prior to implementing the intervention? If so, highlight it here.

p. 31, line 535: Were you able to match if those patients who did not express concern about the relationship/communication style of the therapist were those patients who attended therapists who were more comfortable adapting to implicit learning and reduce their communication with the patient? I am just wondering if those therapists who found it difficult to limit their communication may not have used implicit learning so much and therefore, these patients did also not comment negatively on a particular communication style? I am not sure if you had any measures in place to check if therapists used implicit learning or not – apart from self-report.

p. 32, line 553: the evaluation took part at the end of the trial recruitment phase. What does that mean? Was that before the intervention took place because it was when all sites/participants were recruited? Please clarify.

p. 33 line 580: This is different to what is stated in the online submission form. The authors state that interview transcripts will be made available through the University of Southampton. I am aware this is PLOS One policy, however – would an entire focus group/interview transcript make it easier to identify a site/individual?

Reviewer #2: Review comments:

PONE-D-22-24727: Understanding implementation of a complex intervention in a stroke rehabilitation research trial: a qualitative evaluation using Normalisation Process Theory.

Overall comment: Thank you for the opportunity to review this manuscript. Evaluation of the implementation of complex interventions is an essential element of understanding what factors impact on the effectiveness of novel interventions; publications detailing these factors can be valuable for a wide range of researchers. Normalisation process theory (NPT) is widely utilised to underpin evaluations of the implementation of complex interventions but publications articulating how NPT was used by researchers are less common so it was encouraging to see this manuscript. For me, there was much of value in the use of NPT in this study to evaluate the implementation of what seems to be a novel form of providing therapy in stroke rehabilitation units. My main concerns are firstly, the lack of detail in the description of the intervention being evaluated which made it difficult to assess the quality of the evaluation undertaken, and secondly, the limited way in which the stroke survivor data was utilised in the manuscript. I appreciate that the authors indicate a separate paper will report in more detail on the interview data. However, as the manuscript stands, for me, it does not fully address the aims identified in the abstract and background.

Specific comments are listed based on the section of the manuscript. I have used a lower case m to indicate minor concerns and an upper case M to indicate more major concerns.

Abstract:

Title: The authors may wish to consider including the term process evaluation in the title, this to facilitate location of the publication by search engines. But, see later comments re the use of the term process evaluation in this study and manuscript. (m)

Methods: The number of stroke units participating in the pilot cCRT should be included in the methods section together with a statement that these units were in the UK/England and whether the units were in a single county or region. Similarly, the number of stroke survivors interviewed should be stated. (M)

Results: This section reports on therapists’ perceptions and actions with no indication of findings from data generated from interviews stroke survivors. The same can be said of the conclusions and contributions to the literature section. If the stroke survivor data contributed to the interpretation of the therapists’ data as part of the process evaluation, then some mention of what those data indicated should be made. (M)

Conclusions: Please see the comments which follow relating to determining flexibility and rule based cultures.

Contributions to the literature: May need some revision based on changes suggested as being necessary for the manuscript to be considered for publication.

Main text:

Introduction:

1) Lines 67-74. I appreciate the need for brevity but there needs to be some more detail provided in the description of the intervention being piloted. As a minimum, implicit learning as it was defined in the pilot cCRT needs to be explained, this in order to understand how far the data collection methods selected were likely to be effective in enabling understanding participants’ understanding of the intervention and of the implementation of the intervention. Table 3 (page 17) is very helpful but would be easier to interpret with a more detailed description of the nature of the implicit learning intervention.(M)

2) Lines 91 and 92. The statement ‘here we use implementation science’ is somewhat sweeping. It seems that you are using the terms implementation science and NPT interchangeably. Greater clarity is needed in defining implementation science and your rationale for selecting NPT to underpin the evaluation of implementation in the pilot cCRT. Also, please see comment 5 below re process evaluation.(M)

3) Line 102. Do you mean principles here? (m)

4) Line 100 to 107. Can some indication of content of the 2.5 hour training and perhaps some extracts from the handbook be included as supplementary online material to help readers make sense of the intervention and its evaluation? (m)

5) Line 108. This appears to be the only mention of the term process evaluation. As a minimum this methodology should be explained, ideally in the context of the recent MRC Framework for Developing and Implementing Complex Interventions (Skivington et al, 2021) and the MRC Guidance for Process Evaluation of Complex Interventions (Moore et al (2015). This would set up comment on the choice of qualitative rather than mixed methods in the process evaluation which should be explained here. Generating data as part of a process evaluation on stroke survivor views is most appropriate in my view but please see and consider the above comment relating to the absence of report on how these data contributed to the later interpretation of the therapists’ data.(M)

6) Line 117 and 118. Please clarify the process used for selection of the therapists and therapy assistants. Lines 128-130 suggest a convenience sample. (m)

7) Lines 135-150. The description of the process of analysis in focus group data is helpful overall but the term semantic codes probably needs explanation and a reference, some readers will not be familiar with the approach. Similarly, I suggest you define latent codes, line 146-147. Providing an example (in a table or text box) of the approach reportedly used to integrate the semantic codes and the NPT driven coding (Lines 142-144) would be very helpful.(M)

8) Line 155 and 159. Interviews with in-patients in the hospital setting can be very different to those conducted in peoples own homes, what consideration was given to the impact of setting on data generated? Presumably the lead researcher also conducted the interviews. What adaptations, if any, were needed to allow the stroke survivor with language difficulties to participate in the interview?(M)

9) Line 165 Typo, ‘and’.

10) Lines 165-166. The process of reviewing the focus group data against the stroke survivor interview data needs more explanation, this is potentially a complex process but the information provided in the manuscript is insufficient to determine the quality of the analytical processes used; this in turn impacts the confidence which can be placed on the claims made later about integration of these data. It is not clear why data from control group stroke survivors were included as against only intervention group survivors, brief explanation would be beneficial.(M)

11) Lines 277-291. The term’ handling’ used in the context of post-stroke physiotherapy will be familiar to those working in the field but probably needs definition (as a footnote perhaps) for those not familiar with physiotherapy/occupational therapy or stroke rehabilitation.(m)

12) Line 438 and elsewhere in the reporting of the findings the differentiation of the teams using the concepts of fixed or rule-based and more flexible or principle based makes sense and is indirectly supported by some of the direct quotations and accompanying explanation. What is not clear is out of the four teams how many fell into each category?(M)

13) Across the reporting of findings. It was encouraging to read the stroke survivor perceptions/quotations in relation to NPTs coherence and cognitive participation. It would be helpful to hear comment on why the stroke survivors’ comments did not also relate to collective action and reflexive monitoring.(m)

14) Lines 477-483. Some interesting issues raised here. Firstly, in terms of process evaluation, fidelity to the trial protocol or an intervention manual is commonly evaluated both qualitatively and qualitatively. It seems in this study the two elements were regarded as separate forms of evaluation and the opportunity to consider and synthesise the findings was not taken; for example comparing fidelity measure outcomes for fixed teams versus flexible teams. Is that the case? Secondly, if ‘trial designs (or trials teams) ‘may need to actively facilitate implementation, particularly monitoring and supporting coherence and cognitive participation’, how do the authors envisage this could occur in a trial context? (M)

15) Line 492. Typo ‘e’.

16) Lines 549-553. The criticism (of NPT) is largely valid but is increasingly being addressed in the literature, not least by May et al. Whilst it is encouraging that stroke survivor interviews were included in the qualitative evaluation, the use of the findings from the interview is very limited in this manuscript which in my view limits the confidence that can be placed in some of the claims made. (M)

17) Re limitations, given that there is no indication of the types of units/services and whether they were from one county or across the UK a limitation may relate to their similarity or difference. Some comment should be made regarding the context of rehabilitation service provision in the participating units. (M)

18) Lines 567-569. Some comment on how flexible cultures might be determined and linked to that how change and service improvement might occur in inflexible cultures would be valuable if the findings of this study are to be of value to other researchers. (M)

6. PLOS authors have the option to publish the peer review history of their article (what does this mean?). If published, this will include your full peer review and any attached files.

Reviewer #1: **Yes: **Verena Schadewaldt

Reviewer #2: **Yes: **David J Clarke

---

## [Author Response · Author response to Decision Letter 0]

26 Jan 2023

Thank you for allowing us to revise and resubmit this manuscript. Apologies for missing the deadline - this is due to leave and sickness over the Christmas period. We are pleased to attach a detailed rebuttal letter with this re-submission. If you require any further clarification, please get in touch.

Many thanks

Louise

---

## [Decision Letter · Decision Letter 1]

20 Feb 2023

Understanding implementation of a complex intervention in a stroke rehabilitation research trial: a qualitative evaluation using Normalisation Process Theory.

PONE-D-22-24727R1

Dear Dr. Johnson,

We’re pleased to inform you that your manuscript has been judged scientifically suitable for publication and will be formally accepted for publication once it meets all outstanding technical requirements.

Kind regards,

Adetayo Olorunlana, Ph.D.

Academic Editor

PLOS ONE

Additional Editor Comments (optional):

Reviewers' comments:

Reviewer's Responses to Questions

**Comments to the Author**

1. If the authors have adequately addressed your comments raised in a previous round of review and you feel that this manuscript is now acceptable for publication, you may indicate that here to bypass the “Comments to the Author” section, enter your conflict of interest statement in the “Confidential to Editor” section, and submit your "Accept" recommendation.

Reviewer #1: All comments have been addressed

Reviewer #2: (No Response)

2. Is the manuscript technically sound, and do the data support the conclusions?

Reviewer #1: Yes

Reviewer #2: Yes

3. Has the statistical analysis been performed appropriately and rigorously? 

Reviewer #1: N/A

Reviewer #2: N/A

4. Have the authors made all data underlying the findings in their manuscript fully available?

Reviewer #1: Yes

Reviewer #2: Yes

5. Is the manuscript presented in an intelligible fashion and written in standard English?

Reviewer #1: Yes

Reviewer #2: Yes

6. Review Comments to the Author

Reviewer #1: Thank you for letting me review the revised version of the manuscript. All comments have been addressed.

Reviewer #2: Thank you for the considered responses to the comments and recommendations made. I think the additional information now included in the manuscript, particularly the additional reporting of stroke survivors' experiences, enhances the manuscript. I think reporting of the use of NPT in this study will be useful to other researchers in the field.

7. PLOS authors have the option to publish the peer review history of their article (what does this mean?). If published, this will include your full peer review and any attached files.

Reviewer #1: **Yes: **Verena Schadewaldt

Reviewer #2: **Yes: **David J Clarke

---

## [Editor Report · Acceptance letter]

1 Mar 2023

PONE-D-22-24727R1 

Understanding implementation of a complex intervention in a stroke rehabilitation research trial: a qualitative evaluation using Normalisation Process Theory. 

Dear Dr. Johnson:

I'm pleased to inform you that your manuscript has been deemed suitable for publication in PLOS ONE. Congratulations! Your manuscript is now with our production department. 

Kind regards, 

on behalf of

Associate Professor Adetayo Olorunlana 

Academic Editor

PLOS ONE